# Explainable Model Using Shapley Additive Explanations Approach on Wound Infection after Wide Soft Tissue Sarcoma Resection: “Big Data” Analysis Based on Health Insurance Review and Assessment Service Hub

**DOI:** 10.3390/medicina60020327

**Published:** 2024-02-14

**Authors:** Ji-Hye Choi, Yumin Choi, Kwang-Sig Lee, Ki-Hoon Ahn, Woo Young Jang

**Affiliations:** 1Department of Orthopedic Surgery, Anam Hospital, Korea University College of Medicine, 73 Goryeodae-ro, Seongbuk-gu, Seoul 02841, Republic of Korea; iammeddukk@gmail.com; 2Anam Hospital Bloodless Medicine Center, Korea University College of Medicine, Seoul 02841, Republic of Korea; 3School of Mechanical Engineering, Korea University College of Medicine, 73 Goryeodae-ro, Seongbuk-gu, Seoul 02841, Republic of Korea; ytooom018@naver.com; 4AI Center, Anam Hospital, Korea University College of Medicine, 73 Goryeodae-ro, Seongbuk-gu, Seoul 02841, Republic of Korea; ecophy@hanmail.net; 5Department of Obstetrics and Gynecology, Anam Hospital, Korea University College of Medicine, Seoul 02841, Republic of Korea

**Keywords:** SHAP, Explainable AI, big data, soft tissue sarcoma, perioperative transfusion

## Abstract

*Background and Objectives*: Soft tissue sarcomas represent a heterogeneous group of malignant mesenchymal tissues. Despite their low prevalence, soft tissue sarcomas present clinical challenges for orthopedic surgeons owing to their aggressive nature, and perioperative wound infections. However, the low prevalence of soft tissue sarcomas has hindered the availability of large-scale studies. This study aimed to analyze wound infections after wide resection in patients with soft tissue sarcomas by employing big data analytics from the Hub of the Health Insurance Review and Assessment Service (HIRA). *Materials and Methods*: Patients who underwent wide excision of soft tissue sarcomas between 2010 and 2021 were included. Data were collected from the HIRA database of approximately 50 million individuals’ information in the Republic of Korea. The data collected included demographic information, diagnoses, prescribed medications, and surgical procedures. Random forest has been used to analyze the major associated determinants. A total of 10,906 observations with complete data were divided into training and validation sets in an 80:20 ratio (8773 vs. 2193 cases). Random forest permutation importance was employed to identify the major predictors of infection and Shapley Additive Explanations (SHAP) values were derived to analyze the directions of associations with predictors. *Results*: A total of 10,969 patients who underwent wide excision of soft tissue sarcomas were included. Among the study population, 886 (8.08%) patients had post-operative infections requiring surgery. The overall transfusion rate for wide excision was 20.67% (2267 patients). Risk factors among the comorbidities of each patient with wound infection were analyzed and dependence plots of individual features were visualized. The transfusion dependence plot reveals a distinctive pattern, with SHAP values displaying a negative trend for individuals without blood transfusions and a positive trend for those who received blood transfusions, emphasizing the substantial impact of blood transfusions on the likelihood of wound infection. *Conclusions*: Using the machine learning random forest model and the SHAP values, the perioperative transfusion, male sex, old age, and low SES were important features of wound infection in soft-tissue sarcoma patients.

## 1. Introduction

Soft tissue sarcomas represent a heterogeneous group of rare malignant tumors arising from mesenchymal tissues. Despite their low prevalence, soft tissue sarcomas present a significant clinical challenge for orthopedic surgeons owing to their aggressive nature, propensity for local recurrence, and potential for distant metastasis. The management of soft tissue sarcomas involves a multidisciplinary approach, with surgical resection as the primary treatment modality. Innovation in radiotherapy and systematic treatment as adjuvant therapies are likely to further improve the quality of local control while decreasing the complications and burden of treatment. However, surgery in the soft tissue sarcoma is indicated and remains the best means for control. The success of such interventions is intricately related to the wound-healing process, which plays a pivotal role in preventing complications, enhancing recovery, and improving overall patient outcomes. Postoperative complications in the form of wound-related issues continue to contribute significantly to morbidity, encompassing concerns like wound dehiscence, cellulitis, abscess, seromas, hematomas, and wound necrosis. Factors such as diabetes, smoking, obesity, tumor diameter, and preoperative radiotherapy have been identified as independent predictors for major wound complications. However, existing research is confined to single-institute cohorts, and a comprehensive nationwide study on postoperative wound complications in soft tissue sarcoma patients is lacking [1,2].

Perioperative wound infections present a formidable challenge for clinicians managing patients with sarcoma, as they can lead to prolonged hospital stays, impaired wound healing, compromised oncological outcomes, and increased healthcare costs [3,4]. One factor that warrants further investigation is perioperative blood transfusion [5]. Although the biological mechanisms linking blood transfusion to wound infections are not entirely understood, potential immunomodulatory effects and alterations to immune responses have been proposed as contributing factors.

The low prevalence of soft tissue sarcomas has hindered the availability of large-scale studies focused on wound healing in this specific context. For a single institution to obtain sufficient data to conduct robust investigations is challenging. Consequently, our understanding of the wound healing mechanisms and factors influencing outcomes in patients with soft tissue sarcomas remains limited. Regarding the low prevalence of sarcomas, leveraging the vast repository of information stored in the “big data” Hub of the Health Insurance Review and Assessment Service (HIRA) presents a unique and compelling opportunity to conduct a nationwide analysis of data for sarcoma patients. The HIRA Service Hub is an invaluable resource and data source for diverse healthcare institutions nationwide. By harnessing the power of big data, we can expect to overcome the limitations posed by the rarity of this malignancy and obtain a more comprehensive understanding of wound-healing patterns in sarcoma patients.

When analyzing big data, the sample size can be very large, leading traditional statistical methods to detect even minor differences as statistically significant with high sensitivity. Understanding the complex relationships and contributions of individual features to model predictions is paramount for making informed decisions when dealing with big data. To overcome these limitations, new statistical methods such as machine learning techniques are gaining popularity [6]. Shapley Additive Explanations (SHAP) is a framework used in machine learning for understanding feature importance and to quantify variable contributions to dependent variables [7]. This study approached big data using random forest feature importance and the SHAP framework to analyze the contribution of each feature and interpretable approach to attribute feature importance.

This study aimed to conduct an in-depth analysis of wound infections in patients with soft tissue sarcomas who underwent wide surgical resection. By employing big data analytics from the HIRA Service Hub, our study sought to provide valuable evidence that can pave the way for improved perioperative management strategies, reduced post-operative complications, and favorable treatment outcomes for individuals with this rare and challenging malignancy. This is the first study to identify and analyze variables correlated with wound infection after performing wide resections involving soft tissue sarcomas using a nationwide Korean database. Moreover, the study is the first to analyze feature importance and provide explanations to improve clinical understanding in soft tissue sarcoma using the SHAP approach.

## 2. Materials and Methods

### 2.1. Ethics Approval

This study was approved by the Institutional Review Board of the Tertiary Referral Medical Center, and the need for informed consent was waived owing to the retrospective nature of the study.

### 2.2. Study Populations and Data Source

Data were collected from the HIRA database, which contains information regarding approximately 50 million individuals in the Republic of Korea. The data collected included demographic information, diagnoses, prescribed medications, surgical procedures, and prescription records. Each participant was identified by the unique Korean Resident Registration Number assigned in the Republic of Korea at birth. Data duplication or omission was impossible. All data were anonymized.

The study population consisted of patients who underwent wide excision of soft tissue sarcomas between 2010 and 2021. To recognize patients diagnosed with malignant soft tissue tumors, those assigned to the C49 code (International Statistical Classification of Disease and Related Health Problems, 10th Revision, ICD-10 code) were identified. For patients who underwent wide resection of soft tissue sarcoma, the corresponding Anatomical Therapeutic Chemical (ATC) codes were N0151, NA281, NA282, NA283, NA284, and N0232. After confirming wide resection surgery, patients who underwent infection-related procedures within 3 months after wide resection of soft tissue sarcomas were identified. Patients who received a transfusion during admission for surgery were identified based on ATC codes. Perioperative transfusion was defined as a transfusion performed within 3 months before or after the date of wide resection. All disease data, medication histories, and medical procedures were screened using ICD-10 and ATC codes from the Healthcare Common Procedure Coding System of the HIRA.

### 2.3. Data Analyses

Random forest has been used to predict wound infection after wide resection of soft tissue sarcomas and to analyze the major associated determinants, including transfusion [8,9]. A decision tree is composed of an intermediate node (the test of a predictor), a branch (the value of the predictor as an outcome of the test), and a terminal node (the value of the dependent variable). These trees form a random forest, also referred to as “bootstrap aggregation.” In other words, decision trees are constructed from random samples with replacement (bootstrapping), and they make the majority of the decision on the dependent variable (aggregation) [8]. A total of 10,906 observations with complete data were divided into training and validation sets in an 80:20 ratio (8773 vs. 2193 cases). A standard for the validation of the trained models was the area under the receiver operating characteristic curve (AUC), that is, the area under the plot of sensitivity vs. 1—specificity, which can be considered the degree of sensitivity when its threshold and specificity increase from 0 to 1. Random forest permutation importance was employed to identify the major predictors of infection and SHAP values were derived to analyze the directions of associations with predictors. The random-forest permutation importance calculates the overall decrease in accuracy from the permutation of the data on the predictor. Additionally, the random-forest permutation importance is the average or sum of all trees in a random forest with a range of zero and the number of all trees. The SHAP value of a predictor for a participant is calculated as the difference between what the random forest predicts for the probability of wound infection with and without the predictor. In a hypothetical example of the SHAP values of transfusion for wound infection over the range of −0.02, 0.10, some participants have SHAP values as low as −0.02, and other participants have values as high as 0.10. The inclusion of a predictor (transfusion) into the random forest will decrease or increase the probability of the dependent variable (wound infection) over the range of −0.02 and 0.10. The SHAP values are skewed toward their maximum, hence a positive association exists between transfusion and wound infection in general. Finally, R-Studio 1.3.959 (R-Studio Inc.: Boston, MA, USA) was employed for the analysis between 1 January 2023, and 31 May 2023.

## 3. Results

A total of 10,969 patients who underwent wide excision of soft tissue sarcomas were included. Among the study population, 886 (8.08%) patients had post-operative infections requiring surgery. The overall transfusion rate for wide excision was 20.67% (2267 patients). Risk factors among the comorbidities of each patient with wound infection were analyzed (Table 1).

### SHAP Values

The relative contribution of each comorbidity to wound infection after wide resection of soft tissue sarcoma was compared. Mean SHAP values were calculated to explain and compare the effects of these features. The summary plot displays SHAP values on the horizontal (*X*) axis which represents the average contribution of the feature value to the output. A SHAP value < 0 represents a negative contribution, whereas a value > 0 indicates a positive contribution to the output. A positive contribution indicates that the features were highly important to the outcome. The vertical axis on the left displays features arranged from top to bottom according to their importance. The vertical axis on the right illustrates the values of the features in color: red for high and blue for low. The summary plot demonstrates that transfusion had a highly positive impact on wound infection (Figure 1).

To explain each feature in detail, dependence plots of individual features were visualized.

In the context of the transfusion dependence plot, a noteworthy fact is that SHAP values on the *y* axis exhibited a negative trend among individuals who did not undergo blood transfusions (*x* axis value 0.0), whereas a positive trend was observed on the *Y*-axis among those who received blood transfusions (*x* axis value 1.0). This divergence in SHAP values strongly suggests the significance of blood transfusions as a contributing factor to the likelihood of wound infection. Moreover, an interesting correlation emerged, as we observed that advanced age (denoted by red dots in the plot) corresponded to high SHAP values. This implies that old individuals exhibit an increased susceptibility to wound infections in this context (Figure 2).

In the sex-dependence plot, the 0 and 1 values on the horizontal axis depict males and females, respectively. This finding indicates that males have higher wound infection rates compared to females (Figure 3).

In the age-dependent plot, the SHAP values tended to increase slightly with age. This indicates that the incidence of wound infection increases with age. The right *Y*-axis displays the correlation with transfusions. According to this plot, a large number of people < 20 years of age received a transfusion, and as age increased, the SHAP values for wound infections also increased. Simultaneously, when comparing the *Y*-axis of the same age, an individual who received a transfusion, displayed as a red dot, had a larger SHAP value (Figure 4).

In the socioeconomic status dependence plot, the lower the socioeconomic status (SES), the greater the SHAP value, indicating a high wound infection rate (Figure 5).

## 4. Discussion

This study demonstrated that perioperative transfusion, male sex, old age, and low SES increased the risk of post-operative wound infection based on SHAP values. This nationwide cohort study established a predictive model for post-operative wound infections involving soft tissue sarcomas. The AUC of the prediction model using random forest was 0.6422. This study model has classification ability, but the performance of the study is poor.

The relationship between wound outcomes in soft tissue sarcomas and transfusions remains an intriguing area of investigation in the field of oncology. Although studies have explored the potential impact of transfusions on post-operative wound healing and overall prognosis in cancer patients, any specific link to soft tissue sarcoma outcomes has not yet been fully elucidated. Transfusions, particularly red blood cell transfusions, are often administered to mitigate anemia in cancer patients undergoing surgery or aggressive treatment. However, certain studies have suggested that excessive transfusions may be associated with an increased risk of complications, including surgical site infections, which can be a critical determinant of wound outcomes in patients with soft tissue sarcomas [5]. This study demonstrated that perioperative transfusion was variable with a high predictive value for post-operative wound infection. Although transfusion and post-operative wound infection cannot be taken as proof of a cause-and-effect relationship as the SHAP value of wound infection is higher in the perioperative transfusion variable than in any other single variable, wound infection is expected to be effectively controlled by reducing the risk of transfusion before and after surgery through analysis of SHAP values. Nevertheless, further research is needed to unravel the precise nature of this relationship, considering the diverse histological subtypes and therapeutic approaches for soft tissue sarcomas. A deeper understanding of how transfusion practices affect wound healing and overall prognosis in this context could result in more tailored treatment strategies for these patients.

In males, the feature value for wound infection was the second highest after transfusion. Multiple clinical investigations have indicated sex-based variations in the occurrence of sepsis and its consequences [10]. In a bacteremia epidemiological study, sepsis was more prevalent among males than females [11,12]. Previous studies have presented disadvantages for the male sex due to hormonal differences and an additional high prevalence of malnutrition and comorbidities [13,14,15,16]. Various hypotheses derived from previous studies support our results.

In the age-dependent plot, the SHAP value tended to increase as age increased. As people age, a high chance is present of multiple underlying diseases that can cause problems with wound healing and increase the probability of wound infection [17]. As in Figure 1, liver disease, diabetes mellitus, and other medical issues had a positive impact on the wound infection model output. Although, in the present study, the impacts of each underlying disease on wound infection were analyzed, the effects of an increase in the number of underlying diseases on wound infections were not analyzed.

In terms of SES, the SHAP value was high in patients with low SES, implying that such patients have a high chance of wound infection. The current literature surrounding SES notes that while no universal definition of SES is available, multiple individual SES factors are linked to a heightened risk of compromised wound healing [18,19]. This phenomenon may be attributed to the likelihood of patients with the low SES having a high prevalence in concurrent comorbidities, such as tobacco use, obesity, and diabetes [20]. Furthermore, individuals with lower SES may encounter barriers to accessing essential medical equipment and timely healthcare interventions compared to those with higher SES [21].

In the context of wound infections, factors such as age, sex, and SES exhibit greater importance than specific underlying medical conditions. Additionally, it is of significance that perioperative transfusions have a higher feature importance concerning wound infections when compared to these sociodemographic factors.

Soft tissue sarcomas represent a heterogeneous group of rare malignancies characterized by diverse histological subtypes and anatomical locations. Magnetic Resonance Imaging (MRI) plays a pivotal role in the assessment of soft tissue sarcoma, providing essential imaging features that contribute to the evaluation of treatment strategies, surgical planning, and the prediction of patients’ prognosis [22]. Researchers and clinicians have conducted various studies to assess prognoses in soft tissue sarcoma patients [2,3]. Nonetheless, acknowledging that despite the considerable efforts made to study soft tissue sarcoma wound outcomes, the interpretation of these analyses may be encumbered by a significant limitation is crucial: the low prevalence of sarcoma itself. Soft tissue sarcomas are relatively rare compared with more common malignancies, rendering the sample sizes in many single-institute and multiple-institute studies as relatively small. This inherent rarity poses a challenge in achieving statistical power and generalizability, potentially leading to results that may not accurately reflect the true diversity of this heterogeneous group of cancers. Furthermore, the scarcity of data concerning rare subtypes and specific clinical scenarios of soft tissue sarcomas can further obscure the precision of prognostic models and predictions. This characteristic of sarcomas promotes the use of large databases and population-based nationwide big data registries [23,24,25].

In big data analysis, understanding the complex relationships and contributions of individual features to model predictions is paramount for making informed decisions. As traditional statistical methods encounter limitations in handling large-scale and high-dimensional data, the SHAP framework has emerged as a powerful tool to unravel the intricate dynamics hidden within vast datasets. By leveraging cooperative game theory principles, SHAP provides a comprehensive and interpretable approach to attribute feature importance, even amid the complexities of big data. Predictive modeling encompasses the utilization of data to train machine/deep learning models, enabling the anticipation of future outcomes based on unforeseen data. In the realm of healthcare, predictive modeling assumes a pivotal role, offering insights into diverse areas such as forecasting disease progression, identifying patients susceptible to specific conditions, and optimizing treatment plans. The intricacy of healthcare data and the substantial volume involved pose challenges in healthcare predictive modeling. Machine learning algorithms emerge as potent tools for constructing high-accuracy predictive models, especially in the analysis of clinical and biological data. In the medical domain, a paramount consideration is trustworthiness, reflecting the model’s validity and reliability. The trustworthiness of Artificial Intelligence (AI) is intrinsically linked to the interpretability of the model, addressing the critical question of how individuals can trust AI-generated information when the outcome lacks interpretability. Addressing the interpretability concern involves various strategies, including the development of simplified models through techniques like knowledge distillation. These models preserve high performance while enhancing interpretability. Another approach employs algorithms such as SHAP (SHapley Additive exPlanations) values, which delineate the impact of each feature on a model’s prediction, facilitating a clearer understanding of how different features contribute to the overall result [7,8,9,25,26].

### Limitations

This study had certain limitations. First, this was a retrospective cohort study, and inherent potential bias cannot be ruled out. However, previous studies have validated the accuracy of the HIRA coding system as 70–90%, indicating an acceptable level of accuracy for analyses. Second, comorbidities were identified based on codes. If the diagnosis code was not entered, the possibility of data exclusion existed. Third, the data quality was limited. Nationwide data were obtained through the HIRA, but variables previously identified to be associated with wound infection, such as body weight and operative time, could not be obtained. Social histories, such as smoking, could not be obtained, and no continuous data, such as laboratory results, were present. Finally, Korea is a homogeneous country with few racial variables. Homogeneity can reduce potential biases related to racial or ethnic disparities in terms of access to healthcare, treatments, and outcomes. Researchers could also focus on investigating other potential sources of bias, resulting in more accurate and unbiased results. Although homogeneity can offer advantages in some research contexts, it may also limit the ability to generalize the findings to more diverse populations globally.

Despite the above limitations, this is the first retrospective cohort study of sarcomas conducted using nationwide data. Additionally, this study sought to present a predictive model for post-operative wound infection in soft tissue sarcomas using a random forest model and SHAP values. This explainable model will become the upcoming majority in predicting and analyzing big data.

## 5. Conclusions

The limited prevalence of soft tissue sarcomas has posed challenges for conducting extensive studies. This study aimed to conduct an in-depth analysis of wound infections in patients with soft tissue sarcomas by employing big data analytics from the nationwide database. Employing the machine learning random forest model and the SHAP values, this study identifies perioperative transfusion, male gender, advanced age, and low socioeconomic status (SES) as pivotal factors influencing wound infections in individuals with soft tissue sarcomas.

The investigation gains significance in the context of a global blood shortage following the coronavirus pandemic, elevating the importance of comprehending the necessity and potential complications associated with perioperative transfusions. While infection has previously been suggested as a transfusion-related complication, the lack of substantial data on postoperative wound infections in soft tissue sarcoma underscores the valuable contribution of this study to the existing knowledge in the field.

As for further exploration, future studies could concentrate on gathering and analyzing data to the tumor itself and its treatment. Factors such as size, location, histologic type, chemotherapy, and radiotherapy, considered crucial features in wound infections according to the nationwide database analyzed with SHAP values, could be investigated in detail using hospital-based data.

## Figures and Tables

**Figure 1 medicina-60-00327-f001:**
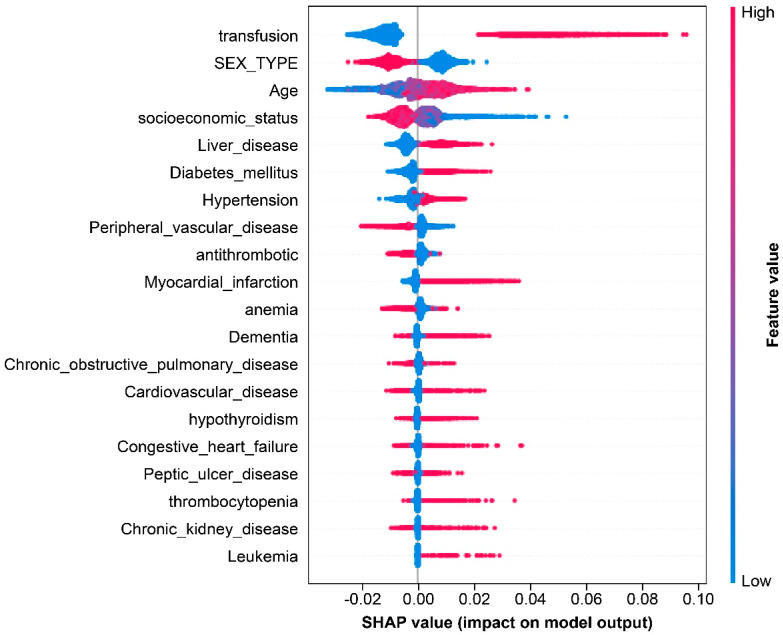
Summary plot of SHAP value. SHAP: Shapley Additive Explanations.

**Figure 2 medicina-60-00327-f002:**
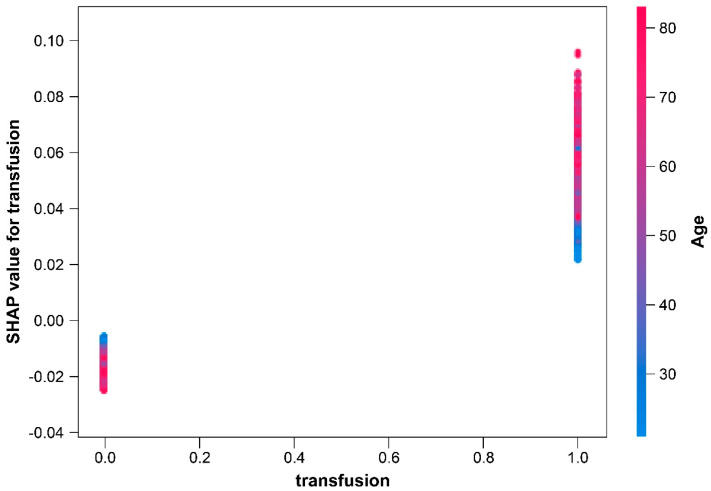
Dependence plot of transfusion.

**Figure 3 medicina-60-00327-f003:**
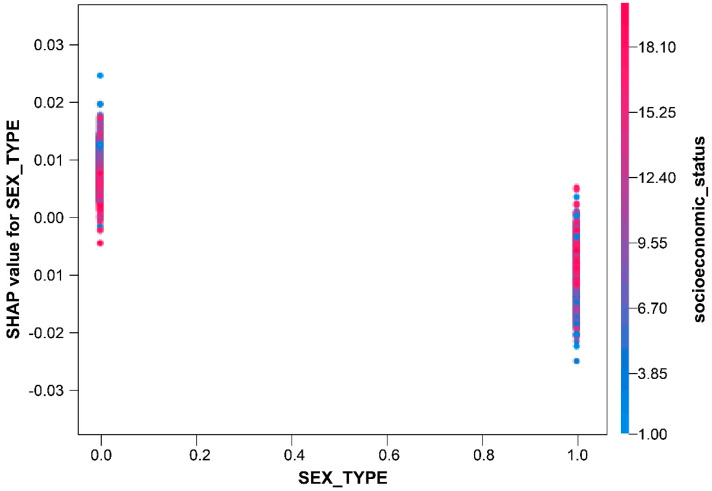
Dependence plot of sex.

**Figure 4 medicina-60-00327-f004:**
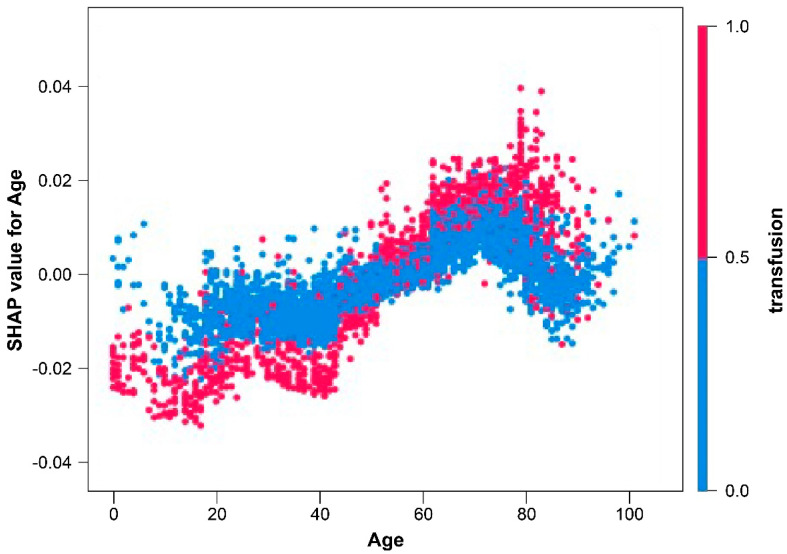
Dependence plot of age.

**Figure 5 medicina-60-00327-f005:**
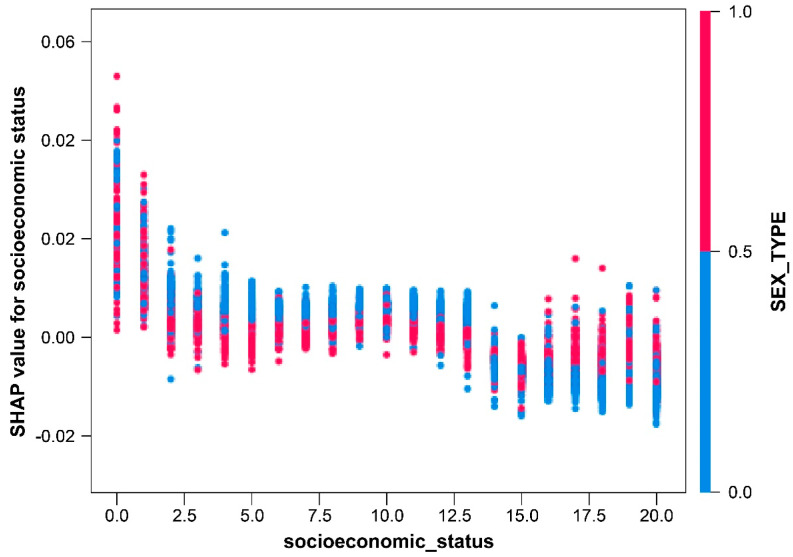
Dependence plot of socioeconomic status.

**Table 1 medicina-60-00327-t001:** Variables and demographic characteristics of the study population.

	Total (*n* = 10,969)		Infection (*n* = 886)		No-Infection (*n* = 10,083)		*p*-Value
	*n*	per	*n*	per	*n*	per	
Age (years)		55.95		58.63		55.71	*p* < 0.05
Transfusion	2267	20.7%	323	36.5%	1944	19.3%	*p* < 0.05
Socioeconomic Status	130,648	11.91	10,209	11.52	120,439	11.94	
Female	4946	45.1%	322	36.3%	4624	45.9%	*p* < 0.05
Liver Disease	3586	32.7%	351	39.6%	3235	32.1%	*p* < 0.05
Iron	805	7.3%	103	11.6%	702	7.0%	*p* < 0.05
Diabetes Mellitus	3051	27.8%	309	34.9%	2742	27.2%	*p* < 0.05
Peripheral Vascular Disease	3097	28.2%	239	27.0%	2858	28.3%	
Hypertension	4443	40.5%	419	47.3%	4024	39.9%	*p* < 0.05
Antithrombotic	7837	71.4%	703	79.3%	7134	70.8%	*p* < 0.05
Anemia	2385	21.7%	215	24.3%	2170	21.5%	
COPD	1896	17.3%	165	18.6%	1731	17.2%	
Cardiovascular Disease	955	8.7%	78	8.8%	877	8.7%	
Congestive Heart Failure	587	5.4%	56	6.3%	531	5.3%	
Peptic Ulcer Disease	1123	10.2%	93	10.5%	1030	10.2%	
Dementia	694	6.3%	68	7.7%	626	6.2%	
Myocardial Infraction	547	5.0%	61	6.9%	486	4.8%	*p* < 0.05
Tranexamic Acid	501	4.6%	51	5.8%	450	4.5%	
Hypothyroidism	626	5.7%	52	5.9%	574	5.7%	
Thrombocytopenia	279	2.5%	31	3.5%	248	2.5%	
Chronic Kidney Disease	257	2.3%	24	2.7%	233	2.3%	
Leukemia	70	0.6%	8	0.9%	62	0.6%	
Thyrotoxicosis Hyperthyroidism	202	1.8%	12	1.4%	190	1.9%	
Hemiplegia	109	1.0%	9	1.0%	100	1.0%	
Solid Tumor	10,790	98.4%	878	99.1%	9912	98.3%	
Lymphoma	156	1.4%	7	0.8%	149	1.5%	
Connective Tissue Disease	111	1.0%	6	0.7%	105	1.0%	
AIDS	20	0.2%	2	0.2%	18	0.2%	

COPD: Chronic Obstructive Pulmonary Disease; AIDS: acquired immunodeficiency syndrome.

## Data Availability

The data used in this study are available from the Big Data Hub of the Health Insurance Review and Assessment (HIRA) Service. However, the data are available for institutions under license and are therefore not publicly available. Access to the study data, however, is available from the authors for researchers who meet the criteria for access to confidential data with the permission of the big data Hub of the HIRA service.

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
