# Peer review of "Explainable Model Using Shapley Additive Explanations Approach on Wound Infection after Wide Soft Tissue Sarcoma Resection: “Big Data” Analysis Based on Health Insurance Review and Assessment Service Hub"

_medicina, 2024, doi:10.3390/medicina60020327_

Round 1
Reviewer 1 Report
Comments and Suggestions for Authors
The manuscript offers a comprehensive analysis of wound infections following wide resection in patients with soft tissue sarcomas, utilizing big data analytics from the Health Insurance Review and Assessment Service Hub in Korea. I found the manuscript to be quite interesting and believe that the research findings have the potential to enhance perioperative management strategies and treatment outcomes for individuals with this challenging malignancy. I do have a few suggestions to enhance the overall quality of the manuscript.
1. In the "Introduction" section, lines 56-57, it would be beneficial to provide a brief expansion on the significance of surgical resection and emphasize its profound impact on patient outcomes.
2. It would be beneficial to provide a concise elaboration on the current understanding of wound healing in patients with soft tissue sarcomas in the "Introduction" section.
3. Most of the references are not up-to-date. It is important to use the most recently published data to provide a background for the research and to discuss the obtained results.
4. In the "Discussion" section, it would be helpful to discuss the challenges and considerations when using AI-based models in biomedical applications. For instance, one of the most important considerations is trustworthiness. How can SHapley Additive exPlanations (SHAP) values address the issues regarding the validity and reliability of a model? (For the reference: https://www.mdpi.com/2313-7673/8/5/442)
5. The "Conclusions" section is poorly written. This section should serve as a concise summary of the main findings and their implications from your study. To improve it, begin by restating the research question or objective to remind readers of the purpose of your research. Next, summarize the key findings and interpret them by discussing their implications and how they contribute to the existing knowledge in the field. Additionally, it would be helpful to suggest future research directions based on your findings, highlighting areas where further investigation is needed.
6. During my review, I noticed a few areas where the language and grammar could benefit from further improvement to enhance the clarity and impact of your research. To ensure the manuscript meets the highest standards of English language usage, I would like to suggest considering the assistance of a professional editor.
Comments on the Quality of English Language
During my review, I noticed a few areas where the language and grammar could benefit from further improvement to enhance the clarity and impact of your research. To ensure the manuscript meets the highest standards of English language usage, I would like to suggest considering the assistance of a professional editor.
Author Response
- In the "Introduction" section, lines 56-57, it would be beneficial to provide a brief expansion on the significance of surgical resection and emphasize its profound impact on patient outcomes.
Thank you for your thorough review of this paper. We have added a sentence and a reference that provide significance of surgical resection on soft tissue sarcoma patient outcomes. The revised introduction section is as below:
The management of soft tissue sarcomas involves a multidisciplinary approach, with surgical resection as the primary treatment modality. Innovation in radiotherapy and systematic treatment as adjuvant therapies, is likely to further improve the quality of local control while decreasing the complications and burden of treatment. However, surgery in the soft tissue sarcoma is indicated and remains the best means for control.
- It would be beneficial to provide a concise elaboration on the current understanding of wound healing in patients with soft tissue sarcomas in the "Introduction" section.
Thank you for your thorough review of this paper. We have added few sentences to provide a concise elaboration on the current understanding of wound healing in patients with soft tissue sarcomas in the Introduction section. The revised introduction section is as below:
The success of such interventions is intricately related to the wound-healing process, which plays a pivotal role in preventing complications, enhancing recovery, and improving overall patient outcomes. Postoperative complications in the form of wound-related issues continue to contribute significantly to morbidity, encompassing concerns like wound dehiscence, cellulitis, abscess, seromas, hematomas, and wound necrosis. Factors such as diabetes, smoking, obesity, tumor diameter, and preoperative radiotherapy have been identified as independent predictors for major wound complications. However, existing research is confined to single-institute cohorts, and a comprehensive nationwide study on postoperative wound complications in soft tissue sarcoma patients is lacking.
- Most of the references are not up-to-date. It is important to use the most recently published data to provide a background for the research and to discuss the obtained results.
We appreciate your comment. We noticed that a few of the references are outdated. References before year 2000 have been revised as possible based on recently published papers.
- In the "Discussion" section, it would be helpful to discuss the challenges and considerations when using AI-based models in biomedical applications. For instance, one of the most important considerations is trustworthiness. How can SHapley Additive exPlanations (SHAP) values address the issues regarding the validity and reliability of a model? (For the reference: https://www.mdpi.com/2313-7673/8/5/442)
Thank you for your thorough review, and important consideration. We have added a paragraph and a reference in the discussion section that discuss the challenges and considerations when using AI-based models, specially SHAP values in this study.
In big data analysis, understanding the complex relationships and contributions of individual features to model predictions is paramount for making informed decisions. As traditional statistical methods encounter limitations in handling large-scale and high-dimensional data, the SHAP framework has emerged as a powerful tool to unravel the intricate dynamics hidden within vast datasets. By leveraging cooperative game theory principles, SHAP provides a comprehensive and interpretable approach to attribute feature importance, even amid the complexities of big data. Predictive modeling encompasses the utilization of data to train machine/deep learning models, enabling the anticipation of future outcomes based on unforeseen data. In the realm of healthcare, predictive modeling assumes a pivotal role, offering insights into diverse areas such as forecasting disease progression, identifying patients susceptible to specific conditions, and optimizing treatment plans. The intricacy of healthcare data and the substantial volume involved pose challenges in healthcare predictive modeling. Machine learning algorithms emerge as potent tools for constructing high-accuracy predictive models, especially in the analysis of clinical and biological data. In the medical domain, a paramount consideration is trustworthiness, reflecting the model's validity and reliability. The trustworthiness of Artificial Intelligence (AI) is intrinsically linked to the interpretability of the model, addressing the critical question of how individuals can trust AI-generated information when the outcome lacks interpretability. Addressing the interpretability concern involves various strategies, including the development of simplified models through techniques like knowledge distillation. These models preserve high performance while enhancing interpretability. Another approach employs algorithms such as SHAP (SHapley Additive exPlanations) values, which delineate the impact of each feature on a model's prediction, facilitating a clearer understanding of how different features contribute to the overall result.
- The "Conclusions" section is poorly written. This section should serve as a concise summary of the main findings and their implications from your study. To improve it, begin by restating the research question or objective to remind readers of the purpose of your research. Next, summarize the key findings and interpret them by discussing their implications and how they contribute to the existing knowledge in the field. Additionally, it would be helpful to suggest future research directions based on your findings, highlighting areas where further investigation is needed.
Thank you for your thorough review of this paper. We have revised the conclusion section that now begins with the restating the research question, followed by summarization of the key findings and contribution to field and suggesting further investigations. The revised conclusion section starts as below:
The limited prevalence of soft tissue sarcomas has posed challenges for conducting extensive studies. This study aimed to conduct an in-depth analysis of wound infections in patients with soft tissue sarcomas by employing big data analytics from the nationwide database. Employing the machine learning random forest model and the SHAP values, this study identifies perioperative transfusion, male gender, advanced age, and low socioeconomic status (SES) as pivotal factors influencing wound infections in individuals with soft tissue sarcomas.
The investigation gains significance in the context of a global blood shortage following the coronavirus pandemic, elevating the importance of comprehending the necessity and potential complications associated with perioperative transfusions. While infection has previously been suggested as a transfusion-related complication, the lack of substantial data on postoperative wound infections in soft tissue sarcoma underscores the valuable contribution of this study to the existing knowledge in the field.
As for further exploration, future studies could concentrate on gathering and analyzing data to the tumor itself and its treatment. Factors such as size, location, histologic type, chemotherapy, and radiotherapy, considered crucial features in wound infections according to the nationwide database analyzed with SHAP values, could be investigated in detail using hospital-based data.
- During my review, I noticed a few areas where the language and grammar could benefit from further improvement to enhance the clarity and impact of your research. To ensure the manuscript meets the highest standards of English language usage, I would like to suggest considering the assistance of a professional editor.
We appreciate your comment. This manuscript was revised from Editage Online.
Reviewer 2 Report
Comments and Suggestions for Authors
This study aimed to analyze wound infections after wide resection in patients with soft tissue sarcomas by employing big data analytics from the HIRA Service Hub.
- Abstract: Results are quite short, please extend.
- Table 1: If possible, please indicate the exact p-values and only use < if under 0.01
- Discussion: Please add a short paragraph about the value of MRI in soft tissue sarcoma diagnostics. Please add the following two papers: https://doi.org/10.1016/j.suronc.2020.08.023
- Conclusion: Your conclusion is quite short. Please extend a little bit.
- Please check for typos. There are some in your manuscript.
Comments on the Quality of English Language
Needs some editing.
Author Response
- Abstract: Results are quite short, please extend.
Thank you for your thorough review of this paper. The result part of the abstract were extended as below.
A total of 10,969 patients who underwent wide excision of soft tissue sarcomas were included. Among the study population, 886 (8.08%) patients had post-operative infections requiring surgery. The overall transfusion rate for wide excision was 20.67% (2,267 patients). Risk factors among the comorbidities of each patient with wound infection were analyzed and dependence plots of individual features were visualized. The transfusion dependence plot reveals a distinctive pattern, with SHAP values displaying a negative trend for individuals without blood transfusions and a positive trend for those who received blood transfusions, emphasizing the substantial impact of blood transfusions on the likelihood of wound infection.
- Table 1: If possible, please indicate the exact p-values and only use < if under 0.01
Thank you for your thorough review of this paper. We set the data analysis of significance level at 0.05 for concern of test power.
- Discussion: Please add a short paragraph about the value of MRI in soft tissue sarcoma diagnostics. Please add the following two papers: https://doi.org/10.1016/j.suronc.2020.08.023
We appreciate your comment. We have added a brief paragraph in the discussion section about the value of MRI in soft tissue sarcoma diagnostics and included a reference according to it.
- Conclusion: Your conclusion is quite short. Please extend a little bit.
Thank you for your thorough review of this paper. We have revised the conclusion section that now begins with the restating the research question, followed by summarization of the key findings and contribution to field and suggesting further investigations. The revised conclusion section starts as below:
The limited prevalence of soft tissue sarcomas has posed challenges for conducting extensive studies. This study aimed to conduct an in-depth analysis of wound infections in patients with soft tissue sarcomas by employing big data analytics from the nationwide database. Employing the machine learning random forest model and the SHAP values, this study identifies perioperative transfusion, male gender, advanced age, and low socioeconomic status (SES) as pivotal factors influencing wound infections in individuals with soft tissue sarcomas.
The investigation gains significance in the context of a global blood shortage following the coronavirus pandemic, elevating the importance of comprehending the necessity and potential complications associated with perioperative transfusions. While infection has previously been suggested as a transfusion-related complication, the lack of substantial data on postoperative wound infections in soft tissue sarcoma underscores the valuable contribution of this study to the existing knowledge in the field.
As for further exploration, future studies could concentrate on gathering and analyzing data to the tumor itself and its treatment. Factors such as size, location, histologic type, chemotherapy, and radiotherapy, considered crucial features in wound infections according to the nationwide database analyzed with SHAP values, could be investigated in detail using hospital-based data.
- Please check for typos. There are some in your manuscript.
We appreciate your comment. Through review throughout the whole manuscript was made.
Reviewer 3 Report
Comments and Suggestions for Authors
#1. As noted by the authors, the variables used in this study are mostly comorbidity data and lack data on the tumor itself and its treatment with respect to size, location, histologic type, chemotherapy, and radiotherapy, which are considered more important. Therefore, it must be said that the usefulness of this study is extremely limited.
#2. The inappropriate use of abbreviations and lack of full spelling in the manuscript, including Abstract, is unkind to the reader. Abbreviations should be accompanied by the full spelling the first time they are used.
Comments on the Quality of English Language
English used in this manuscript is almost acceptable, but minor editing is required.
Author Response
#1. As noted by the authors, the variables used in this study are mostly comorbidity data and lack data on the tumor itself and its treatment with respect to size, location, histologic type, chemotherapy, and radiotherapy, which are considered more important. Therefore, it must be said that the usefulness of this study is extremely limited.
Thank you for your thorough review of this paper. We are aware of the limitations of our study and have revised the conclusion section that suggests further investigations considering that the variables used in this study lack data on the tumor itself. The revised conclusion section starts as below:
As for further exploration, future studies could concentrate on gathering and analyzing data to the tumor itself and its treatment. Factors such as size, location, histologic type, chemotherapy, and radiotherapy, considered crucial features in wound infections according to the nationwide database analyzed with SHAP values, could be investigated in detail using hospital-based data.
#2. The inappropriate use of abbreviations and lack of full spelling in the manuscript, including Abstract, is unkind to the reader. Abbreviations should be accompanied by the full spelling the first time they are used.
Thank you for your thorough review of this paper. We checked throughout the manuscript from the abstract that abbreviations are accompanied by full spellings at first time.
Round 2
Reviewer 3 Report
Comments and Suggestions for Authors
The authors have responded appropriately to the reviewers' comments and have made revisions to the manuscript.
Comments on the Quality of English Language
There are no significant problems with the English of this paper.